# The influence of small farm reservoir network characteristics on their cumulative hydrological impacts

Henri Lechevallier<sup>1, 2</sup>, Cécile Dagès<sup>1</sup>, Delphine Burger-Leenhardt<sup>2</sup>, Claire Magand<sup>3</sup>, and Jérôme Molénat<sup>1</sup>

<sup>1</sup>LISAH, Univ Montpellier, AgroParisTech, INRAE, IRD, Institut Agro, Montpellier, France

Correspondence: Henri Lechevallier (henri.lechevallier@inrae.fr)

### Abstract.

In many regions of the world, the use of infrastructure to store runoff and stream water, such as small farm reservoirs, is the only way to enable irrigation and thereby secure and increase food production. The presence of multiple reservoirs in one catchment has cumulative impacts that are not necessarily the sum of the individual impacts. However, we still have little knowledge of the spatial factors that drive these cumulative impacts. In this work, the effects of the distribution of small reservoirs in a catchment on their hydrological impacts are investigated with a modeling approach. Our numerical experiment consists of randomly generating multiple small reservoir networks in the same catchment with realistic reservoir numbers, capacities, and spatial distributions and then comparing their hydrological impacts over a 20-year period. We focused on two variables, namely, the outlet discharge and the mean proportion of the network in low flow, which we computed annually and seasonally. We used the distributed agrohydrological model MHYDAS-small-reservoir, which represents small reservoirs and their links with the hydrological network and the irrigated plots. In our context and with current reservoir management rules, we found that the impacts of reservoirs are more important in summer, with discharges reduced by more than 20 % and up to 60 % compared with the reference situation without reservoirs. Moreover, low flow proportions are always higher than those in the reference situation. For these two indicators, the main explanatory factors are the number and distribution of reservoirs, with a limited effect of the storage capacity. The effects of the study factors on the seasonal and annual indicators were thoroughly interpreted with respect to the hydrological functioning of the catchment and the timing and amount of irrigation. This work contributes to a better understanding of the drivers of the cumulative hydrological impacts of small reservoirs. Although many questions remain, our results can help scientists and water managers choose the best representation of small reservoirs in their models to address their needs.

### 20 1 Introduction

Agricultural water management during dry periods is a common issue in many regions of the world. Small farm reservoirs are usually considered a solution to secure water resources for irrigation during the dry season, when the availability of surface water in streams is not guaranteed and is subject to restrictions from local authorities (Carluer et al., 2017). They have been

<sup>&</sup>lt;sup>2</sup>G-EAU, Univ Montpellier, AgroParisTech, BRGM, CIRAD, INRAE, IRD, Institut Agro, Montpellier, France

<sup>&</sup>lt;sup>3</sup>OFB, Orléans, France

increasingly built since the 1950s in many parts of the world, such as Brazil (Pinhati et al., 2020), South Africa (Hughes and Mantel, 2010), the USA (Deitch et al., 2013), Australia (Schreider et al., 2002) and France (Galéa et al., 2005). These reservoirs can be placed directly on the stream or inserted in specific locations in the catchment to capture surface runoff and drainage waters (hill reservoirs). There are no common criteria for differentiating small reservoirs from medium or large reservoirs. Depending on the study context (e.g., location in the world or size of the study catchment), they can be defined based on a maximum surface (e.g. in Morden et al., 2022) or a maximum volume (e.g. in Ayalew et al., 2017). In this study, small reservoirs are defined as reservoirs with a capacity of less than  $1 \, Mm^3$ . Small reservoirs can have many uses, such as watering, flood mitigation, or groundwater recharge, but we focus on small farm reservoirs built for irrigation purposes.

Compared with other solutions for the storage of water for irrigation (e.g., storage in large reservoirs with shared use), small reservoirs are believed to have lower individual impacts on river flows and contribute to better equity between local actors in terms of water availability (van der Zaag and Gupta, 2008). However, the impacts of small farm reservoirs accumulate in a catchment along the stream (Habets et al., 2018). Thus, the impacts of a reservoir network cannot be considered as the sum of the individual impact of each reservoir. The main cumulative impact is hydrological, i.e., stream flows are modified by the presence of reservoirs. Hydrological changes can induce other types of impacts, such as ecological, geomorphological, or biogeochemical impacts (Kennon, 1966; O'Connor, 2001; Seyedhashemi et al., 2021). These other impacts may arise not only from changes in the mean stream flow but also from variations in low flows and high flows. In agricultural catchments, low flows are usually a concern, as they can have significant effects on local and downstream stream ecology (Sarremejane et al., 2022). Many attempts have been made to assess the cumulative impacts of small farm reservoirs in different catchments around the world, and methods, especially numerical models, have been developed for this purpose (Habets et al., 2018).

The outlet discharge is the most frequently used variable to evaluate the cumulative impacts of small reservoirs (irrespective of their use). Changes in the outlet discharge are often analyzed annually (e.g. Kennon, 1966; Neal et al., 2002; Hughes and Mantel, 2010; Xu et al., 2013; Yan et al., 2023) and/or monthly or seasonally (e.g. Hughes and Mantel, 2010; Fowler et al., 2015). The impacts of small farm reservoirs on flood and low flows have been less studied. In these cases, the indicators used are mostly based on outlet discharge (e.g. Galéa et al., 2005; Robertson et al., 2023; Xu et al., 2022). Few studies use spatialized approaches to characterize the impacts of reservoirs along the hydrological network (e.g. Güntner et al., 2004; Ayalew et al., 2017; Deitch et al., 2013).

Studies on the hydrological impacts of small farm reservoirs usually aim at quantifying these impacts for the current composition of the catchment in reservoirs and for the current water use (e.g. Kennon, 1966; Nathan et al., 2005; Alcorn, 2007; Deitch et al., 2013) or for some scenarios related to climate (e.g. Krol et al., 2006; Habets et al., 2014), the number of reservoirs (e.g. Meigh, 1995; Rabelo et al., 2022), their capacity (e.g. Rabelo et al., 2021), the timing and amount of withdrawals (e.g. Meigh, 1995; Brasil and Medeiros, 2020), or the filling period of reservoirs (e.g. Habets et al., 2014; Pinhati et al., 2020). However, few studies focus exclusively on understanding the factors that drive the hydrological impacts of small farm reservoirs. A better

understanding is critical for land planners to determine how to minimize the impacts of existing or additional reservoirs in a catchment. The driving factors can be classified into two main categories:

- Characteristics of the reservoir network; capacity, number and spatial distribution of reservoirs.
- Management of reservoirs: filling method, link to the downstream stream, water use intensity and timing.

These factors are closely related to the pedoclimatic context and to the farming system (type of crops and type of practices). These contextual elements usually affect the development of small reservoirs in regulated catchments. A change in climate or in the farming system can trigger the construction of more reservoirs or lead to changes in their management and thereby modify their hydrological impacts.

It is well established that an increase in the total storage capacity (i.e., an increase in the reservoir size or in the number of reservoirs) associated with an increase in water use leads to a decrease in annual outlet discharge (e.g. Savadamuthu, 2002; Teoh, 2003; Thompson, 2012; Habets et al., 2014). However, its effect on other variables (e.g., low flow) remains unclear. Furthermore, the effects of the number of reservoirs with a constant capacity and of the distribution of reservoirs have rarely been studied (rare examples are Ayalew et al., 2015; Meigh, 1995).

In this study, we assume that the characteristics of small reservoir networks are key factors in their cumulative hydrological impacts. Given this assumption, the objective of the study is to quantify and better understand the cumulative hydrological impacts in a catchment of (i) the density of small reservoirs, (ii) the total capacity of the reservoirs, and (iii) the distribution of reservoirs along the stream network. To go beyond previous studies on this topic, we considered the hydrological impacts not only on mean stream flow but also on low flows. These variables were analyzed annually and seasonally. For low flows, the cumulative impact was quantified and analyzed spatially with respect to the stream network rather than solely at the catchment scale. As experimental or observational approaches are not feasible, we adopted a modeling approach using the spatially distributed agrohydrological model MHYDAS-small-reservoir (Lebon et al., 2022). A 20-year numerical experiment was conducted for a catchment both with and without small-reservoir networks with variable characteristics. The Gélon catchment, a typical third-order catchment in southwestern France with an intermittent stream, was selected as the basis for the numerical experiment. With this approach, we were able to evaluate the effects of the three study factors on the mean stream flow and low flow and analyze the processes driving these effects.

100

### 2 Materials and Methods

### 2.1 The model: MHYDAS-small-reservoir

### 2.1.1 Presentation of the model

MHYDAS-small-reservoir is a spatially distributed agro-hydrological model that includes a representation of small reservoirs (Lebon et al., 2022). It is composed of a soil-crop model (Constantin et al., 2015), a groundwater model (Kirchner, 2009), a water routing model (Moussa et al., 2002), a reservoir model (Lebon et al., 2022), and a decision model for farming practices and irrigation (Murgue et al., 2014). It operates at an hourly time step for water routing and a daily time step for crop growth.

The space is described with four types of compartments: agricultural or natural surfaces, on which vegetation is growing, groundwater bodies, the hydrological network, and reservoirs. Each compartment is discretized into calculation units, which are named surface units (SUs), groundwater units (GUs), reach sections (RSs) and reservoirs (REs), respectively. Each groundwater unit corresponds to an independent groundwater body that flows directly into the hydrological network. Surface units follow parcel shapes and topography so that each unit is hydrologically linked to only one other SU, RS or RE and to one GU. The temporal resolution is adapted to each process to be represented and thus varies from 1 hour to 1 day.

An extensive description of the model can be found in Lebon et al. (2022) and Lebon (2021). In the following, we detail how reservoirs are represented in the model and how withdrawals and irrigation are addressed, as this information is essential for the experiment.

### 2.1.2 Representation of small reservoirs in the model

O5 Small reservoirs are represented individually and can be directly connected to the stream network or not (hill reservoirs). They capture all upstream water and spill when they are full. An ecological flow can be fixed for each reservoir: whenever water flows upstream of the reservoir, this flow must be guaranteed downstream. In the numerical experiment, the minimal flow is fixed to 10 % of the mean interannual discharge at the reservoir location, according to French regulations. Evaporation from the reservoir water is considered proportional to the reference evapotranspiration, with a coefficient of 0.6. Percolation from the reservoir bed to groundwater or through the dam wall is not considered in the model. The shape of all reservoirs is the same and corresponds to a reversed half-pyramid, following the relationship reported by Liebe et al. (2005). More information on the relationship between the area and the volume is provided in the Supplementary Material.

### 2.1.3 Withdrawals and irrigation

A decision model (Murgue et al., 2014) simulates the volumes of water withdrawn from small reservoirs and the water applied by irrigation to the crop for each irrigated field depending on the crop demand and the availability of water in each reservoir. There is no constraint on the annual amount of water withdrawn from reservoirs, and reservoirs can fill throughout the year.

**Figure 1.** Localization of the Gélon catchment in France and spatial discretization in MHYDAS-small-reservoirs for the current network of small reservoirs.

However, withdrawals in a reservoir are possible only if the water volume is higher than 1/4 of its capacity. The use of a decision model is one of the specificities of MHYDAS-small-reservoir compared with most other models used to evaluate the impacts of small reservoirs. More information on the decision model for irrigation can be found in the Supplementary Material.

### 2.2 Numerical experiment

### 2.2.1 Support site

The Gélon catchment (Figure 1) is the support site for the numerical experiment. It is a  $19.8\ km^2$  hilly catchment in southwestern France, with soils composed mainly of alluvial and molassic slope deposits (Party et al., 2016) with a clay loam texture. Soil is highly impermeable, which leads to a dense hydrological network with many irregular sources. There is no deep aquifer (Cavaillé, 1968). The climate is temperate without a dry season with a warm summer according to the Köppen-Geiger climate classification (Strohmenger et al., 2024). On average, from 1989 to 2016, the annual rainfall was 675 mm, the annual ET0 was 905 mm, and the mean temperature was  $13.5\ ^{\circ}$ C (Lebon et al., 2022). There are currently 25 reservoirs, with a total estimated capacity of  $205000\ m^3$ , in the catchment.

The field layout, stream network and ratio of agricultural land to noncultivated land at the support site correspond to real-world conditions and are represented in the model as specified in Lebon et al. (2022). Thus, the Gélon catchment is divided in the model into 2402 SUs, representing approximately  $15~\rm km^2$  of agricultural land and  $5~\rm km^2$  of noncultivated land. The  $8~\rm km$  long hydrological network is divided into  $365~\rm RSs$  (Figure 1).

120

125

MHYDAS-small-reservoir was previously applied, calibrated and validated on the Gélon catchment for the hydrological year 2014/2015 to evaluate the impacts of the existing reservoirs (Lebon et al., 2022). Compared with the previous use of the model on the Gélon catchment (Lebon et al. (2022)), the number of groundwater units was increased from 17 to 282 to better fit the field observations. This adjustment did not considerably modify the flows at the outlet, but the discharges in the hydrological network better matched expectations.

In this work, we were not interested in the hydrological impacts of current reservoirs. The catchment served as a basis on which the model accurately represented the processes related to water flows and crop growth and management in the agropedo-climatic context of southwestern France, and we used the model with hypothetical numbers, positions, and characteristics of reservoirs.

### 2.2.2 Approach

150

165

The numerical experiment involved generating 90 situations, each corresponding to a randomly generated reservoir network. Each network had a different reservoir density, total capacity, or spatial distribution. Under real conditions, the irrigated fields are located close to the small reservoir used for irrigation. Therefore, for each generated reservoir network, we also determined a specific spatial allocation of irrigated crops that corresponded to a predefined statistical distribution of irrigated crops in the catchment.

For each study factor, different modalities were chosen prior to the experiment: three for the number of reservoirs, two for total stored capacity in the catchment, and three for the random placement of reservoirs on the hydrological network. Each combination of modalities is repeated 5 times, leading to a total of 3x2x3x5 = 90 situations.

With this approach, two situations have:

- Different or equal numbers of reservoirs depending on the chosen value.
- Different or equal reservoir capacities depending on the chosen values for the total stored capacity and the number of
   reservoirs.
  - Different positions of reservoirs in the hydrological network. Depending on the chosen method, there are more reservoirs upstream, downstream, or they are more equally distributed.
  - Different irrigable parcels depending on the placement of reservoirs. The crops on these parcels are different from those
    in the reference situation. For a parcel selected in two situations, the associated irrigable crop can be different.
  - Equal total surface area of irrigable land and equal total surface area of each irrigable crop.

For each network, a simulation was performed with the MHYDAS-small-reservoir model. In addition, a simulation without any reservoirs or irrigation was performed to serve as the reference situation. The hydrological impact of each generated

**Table 1.** Distribution of irrigable crops used in the numerical experiment and mean irrigation water use for these crops. The distribution is calculated from regional data, and the mean irrigation water use is evaluated with the model with simulations performed over the study period with a single parcel and a single crop without limitations on water availability.

| Crop             | Percent | Mean interannual water use (mm) |  |
|------------------|---------|---------------------------------|--|
| Maize            | 54      | 230                             |  |
| Maize for silage | 5       | 160                             |  |
| Maize for seeds  | 12      | 270                             |  |
| Soybeans         | 12      | 280                             |  |
| Straw cereals    | 15      | 20                              |  |
| Sorghum          | 1       | 50                              |  |
| Colza            | 1       | 12                              |  |

reservoir network was quantified as the difference between the simulation with that network and the reference situation. The situations with reservoirs are designated "impacted situations".

In the following sections, we describe the choice of values for the study factors and the distribution of irrigated crops (Section 2.2.3), the method to generate each of the 90 reservoir networks (Section 2.2.4), and the method for the random allocation of crops near reservoirs (Section 2.2.5). Finally, we detail the setup of the simulations in Section 2.2.6 (simulation period, initialization, and crop and weather data).

### 175 **2.2.3** Values for the study factors

170

Three values were chosen for the total number of reservoirs: 7, 14, and 21. They correspond to densities of approximately 0.35, 0.70, and 1.05 km<sup>-2</sup>, as these values are quite representative of this region (DDT82, 2022). To determine the distribution of reservoirs along the stream, three methods were designed. They are designated "upstream", "balanced", and "downstream" and are described in more detail in Section 2.2.4. As hill reservoirs are usually found in specific locations, a random placement does not make much sense for this type of reservoir. Therefore, the reservoirs are placed only on the hydrological network in the experiment. The total stored volume was fixed along with the total irrigable area and the distribution of irrigable crops. The distribution of irrigable crops was calculated from regional data (i.e. Pignard et al., 2023). The main irrigable crops are maize, straw cereals, and soybeans (Table 1).

The total irrigable area is fixed so that the mean water use for irrigation without limitations represents 5 % of the mean annual naturalized flow determined in the reference situation. The mean water use was estimated with the crop model (i.e. AqYield, Constantin et al., 2015) for each crop individually (Table 1). This estimation led to a surface of approximately 1 km<sup>2</sup>

(the value of  $1~\rm km^2$  was retained) and an annual water need of  $210000~\rm m^3$ . Considering that only 3/4 of the water stocks in reservoirs can be used for irrigation (see Section 2.1.2), this situation led to a value of  $280000~\rm m^3$  for the storage capacity. The value of  $280000~\rm m^3$  thus corresponds to a situation where the total stock in reservoirs at the beginning of the cropping season is sufficient to cover the irrigation demand in average years. The second value tested for the total capacity was fixed to  $140000~\rm m^3$ , representing a situation where water stored in reservoirs in winter alone will probably not be sufficient to cover all the water demand. The chosen values were determined to be reasonable considering the current estimated storage of approximately  $205000~\rm m^3$  distributed into 25 small reservoirs in the Gélon catchment.

### 2.2.4 Creation of the reservoir networks

The starting point for generating each network is the catchment in the reference situation without reservoirs. The placement of small reservoirs on the hydrological network consists of selecting a set of RSs on which the chosen number of reservoirs will be placed. For this purpose, the 365 RSs that make up the hydrological network in the numerical representation of the catchment are divided into two subsets: upstream and downstream. The criterion used to separate the two subsets is a threshold of the drained area. This threshold corresponds to the maximum drained area of first-order streams (approximately 2.5 km²). Since RSs have different lengths, it is useful to compare the sizes of the subsets by their total lengths. A total of 55 % and 45 % of the total network length are included in the upstream and downstream subsets, respectively (see the Supplementary Material for more information). The placement of a reservoir on the network is carried out with two consecutive random draws. In the first draw, one of the subsets is selected, and in the second draw, one of the RSs of the selected subset is chosen. In the first draw, the probability associated with each subset depends on the selected method. For the methods "upstream", "balanced", and "downstream", the probabilities of drawing the upstream subset are 0.8, 0.5, and 0.2, respectively. In the second draw, the selection of each RS of the selected subset is equiprobable.

Once a location is selected for each reservoir, the hydrological network is modified to include the new reservoirs. The total storage capacity is evenly distributed across all the reservoirs, and the area of the neighboring SU is reduced to consider the spatial extent of the reservoirs (see the Supplementary Material for the shape of reservoirs and the relationship of the area to the volume).

# 2.2.5 Random allocation of crops

After the reservoirs are placed on the hydrological network, the allocation of crops is carried out in two steps. First, a set of SUs is randomly selected near each RE (within a distance of 1000 meters) to reach a total of 1 km<sup>2</sup> of irrigated land in the catchment that is evenly distributed between all the reservoirs. A tolerance threshold of 1 % is applied to the value of 1 km<sup>2</sup> to address the different parcel sizes. An irrigable crop is subsequently associated with each of the selected parcels. The irrigable crops are chosen among the predefined set (Table 1), and the distribution between the available parcels is determined to have the same distribution of irrigable crops at the catchment level in all 90 situations with a tolerance threshold of 2 %.

https://doi.org/10.5194/egusphere-2025-4737 Preprint. Discussion started: 15 October 2025

© Author(s) 2025. CC BY 4.0 License.

# 2.2.6 Setup of simulations

The 90 + 1 simulations are run from 1995/09/01 to 2021/01/01. The parametrization of the model is the same as that in Lebon et al. (2022). Since the initial conditions cannot be determined, we use a warm-up period instead. Lebon (2021) reported that for the application of the MHYDAS-small-reservoir to the Gélon catchment, a warm-up period between 2 and 5 years was sufficient to reach satisfactory initial conditions. Here, we consider a 5-year warm-up period. These five years are not included in the analysis.

Each agricultural SU is associated with a crop. Although the model can support crop succession, only one crop was associated with each SU for all the simulated years, corresponding to the main crop of the 2014-2015 cropping season, which is available in the French Land Parcel Identification System (IGN, 2015). Thus, each year can be seen as the repetition of the same agricultural year with varying initial conditions and weather.

The weather data are composed of hourly SAFRAN predictions for rainfall and reference evapotranspiration (Penman-Monteith) and daily predictions for the mean and minimum daily temperatures used in the crop model. The SAFRAN climatic data are provided by Météo-France and were downloaded via the SICLIMA platform developed by AgroClim-INRAE. The data are available in the form of 8x8 km<sup>2</sup> grids (Bertuzzi et al., 2022). The Gélon intersects with 4 of these grids, so the spatial variability of the weather data is considered. The seasonal rainfall for the main cell is presented in Figure 2.

### 2.3 Method of analysis

# 2.3.1 Indicators of impact

To analyze and compare 90+1 simulations, synthetic indicators are needed. In this study, 2 indicators were chosen: the outlet discharge and the proportion of the network in low flow. The outlet discharge is commonly reported in the literature. It is also easy to compute and compare between simulations. However, it provides information only on what happens at the outlet and not on the remaining hydrological network.

The proportion of the network in low flow is an indicator developed in this study to provide information on flow that is spatially aggregated. Each day, the total length of the network in low flow is computed by comparing the discharge in each RS with a local low flow threshold. Afterward, the mean proportion of the network in low flow during the study period (a year, a season, a month) can be computed.

The daily discharges on the 365 RS in the reference situation are used to compute these low flow thresholds for each RS. They correspond to the Q90, the discharge that is exceeded 90 % of the time, computed during the study period.

**Figure 2.** Seasonal rainfall during the study period for SAFRAN cell 8558. Years start in spring (on 1 April). The yearly rainfall height in mm is indicated on top of the bars for each year.

Both the daily outlet discharge and the proportion of the network in low flow are outputs of the model. They are further aggregated for analysis.

### 255 2.3.2 Seasonal and annual aggregation

The results are analyzed on a yearly and seasonal basis. The years of analysis span 1 April of year N to 31 March of year N+1. Thus, the impacts of reservoirs are studied for a period in which they are first emptied in spring and summer and then filled the remainder of the year to reach their maximum capacity at the beginning of each year. This was effectively observed for nearly every year and situation (see Section 3.3.2). In the Results section, the seasons are therefore displayed in the following order: spring (civil year N), summer (N), autumn (N), and winter (N+1). The simulation results are thus analyzed from 2001/04/01 to 2020/03/31, which constitutes a total of 19 years.

# 2.3.3 Steps of analysis

260

265

With this numerical experiment, our goal is to gain knowledge on the following:

- 1. The hydrological impacts of reservoirs, described by the two impact variables.
- 2. The effect of each study factor on the indicators of impact.

- 3. The relative magnitude of the effect of each study factor.
- 4. The why and how of the impacts of small reservoirs and the role of the study factors.

As there is high interannual variability in climate forcing (see Figure 2), the results are presented year by year. Boxplots are constructed to summarize the impacts found in the 90 simulations and to show the effect of each factor. To quantify the relative importance of each factor on each variable (point 3), we use decompositions of variance with a linear model (ANOVA), including interaction terms. Since the effect of a factor on an indicator can differ from year to year, we perform the ANOVAs year by year. For clarity, the outcomes of points 1 to 3 are summarized in Table 2 at the beginning of the discussion.

### 3 Results

280

295

### 3.1 Quantification of reservoir impacts

### 275 3.1.1 Annual impacts

Simulated annual outlet discharges in the reference situation show large interannual variability, with values ranging from 80 mm to 400 mm (e.g., from  $1.6 \times 10^6$   $m^3$  to  $8 \times 10^6$   $m^3$ ). As expected, the outlet discharge is always lower in the impacted situation than in the reference situation (Figure 3a). However, relative to the reference situation, the decreases in outlet discharge are small, and the variability between situations for a given year is quite small, especially compared with the interannual variability in discharge. Absolute decreases in annual outlet discharge are usually between 4 and 9 mm (Figure 7), which represents between 1 and 6 % of the annual outlet discharge in the reference situation, except in 2011 and 2016, when it represents between 5 and 15 % of the annual outlet discharge. These two years are the years with the lowest total rainfall (Figure 2).

Compared with the annual outlet discharge, the impacts of reservoirs on the annual proportion of the network in low flow exhibit high variability between the simulations (Figure 3b). Depending on the year and the situation, the proportion of the network in low flow can increase or decrease compared with the reference. For years with low proportions of the network in low flow in the reference situation (i.e., <20 %), reservoirs usually increase this proportion. In particular, in years with no low flow in the reference situation, such as 2010 or 2013, the proportion of the network in low flow can reach 10 % in impacted situations. For years with high low flow proportions in the reference situation (i.e., >20 %), the effect of reservoirs on low flow can be positive or negative depending on the situation. For these years, the variability between the impacted situations is usually higher.

# 3.1.2 Seasonal impacts

The impact of reservoirs on outlet discharge is more important in summer in both relative and absolute terms. The summer decreases in outlet discharge are consistently higher than 2 mm and can reach 10 mm, with most values in the 3-7 mm range.

300

**Figure 3.** Boxplot of annual outlet discharges (a) and annual proportions of the network in low flow (b) in the 90 situations for the simulated years. X represents the values in the reference situation. The years analyzed span April n to March n+1.

In the other seasons, the discharge usually decreases compared with the reference situation, but there are also some years and some situations for which it increases, especially in autumn. This finding can be explained only by irrigation return flows, which means that these flows can occur after the end of the irrigation period. In autumn, years with increases or decreases in the outlet discharge in the impacted situations are usually the same for all simulations, which highlights the role of weather. For example, 2001, 2002, 2005, 2006, 2011, 2014, 2015 and 2017 are the years in which the autumnal outlet discharge increased, and the summer rainfall of all these years is higher than the median value during that period (Figure 2).

**Figure 4.** Boxplot of absolute (a) and relative (b) changes in the seasonal outlet discharge and boxplot of the seasonal low flow proportion (c) for the simulated years and the 90 impacted situations compared with the reference situation.

In all situations, most of the low flow occurs in summer and in autumn (Figure 4c). In the reference situation, there is no low flow in spring and winter, except for the years 2002 and 2017, when low flow lasts until winter. In spring, the proportion

of the network in low flow remains low for all years in the impacted situations (except in 2011). In summer, the low-flow proportion increases in all the impacted situations, generally by at least 5 % and up to 30 % (in absolute terms). In autumn, the effect of reservoirs depends on the year. Compared with the reference, the proportion of the network in low flow can either increase or decrease, but for most years, it tends in the same direction for all the impacted situations. Except for some years (especially 2001, 2016 and 2017), the variability between the impact situations is lower than that in summer. For years with a high proportion of the network in low flow in autumn in the reference situation (>20 %), the presence of small reservoirs always decreases the proportion. In the winter, the proportion of the network in low flow in the impact situations is usually close to 0. For the two years with extended low flow in winter, the proportion decreases slightly in the impacted situations. In summary, compared with the reference situation, small reservoirs generally (i) increase the annual proportion of the network in low flow (Figure 3b) and (ii) modify the low flow period, which starts earlier, in spring or summer, and can also end earlier in autumn.

### 3.2 Effect of the study factors

To analyze the effects on the hydrological impacts of the different factors, namely, the total storage capacity, density and spatial distribution of reservoirs, we focus on the indicators for which the impact of the reservoirs is the most important and variable. The five indicators we selected are (i) the annual outlet discharge, (ii) the summer outlet discharge, (iii) the annual proportion of the network in low flow, (iv) the summer proportion of the network in low flow, and (v) the annual proportion of the network in low flow. This analysis is presented in two subsections, each corresponding to a different step. First, we analyze in which directions these factors affect the indicators, i.e., whether the factors have a positive or negative impact. Second, we quantify the relative contribution of the three factors to the impact on the indicators. The effect, or influence, of each factor is therefore analyzed in terms of the direction and magnitude.

### 3.2.1 Directions of the effects

Among the five indicators, the storage capacity consistently influences the annual and summer outlet discharge (Figure 5 left). In both cases, increased capacity leads to higher impacts. The effect on annual discharge is clearer than that on summer discharge (the boxes are more separated). For most years, the effect of storage capacity on low flow is unclear. With respect to the annual proportion of the network in low flow, the storage capacity is relevant only in 2001, 2016, and 2017, which were all particularly dry years in terms of rainfall, especially in autumn. In general, higher storage capacities seem to be associated with lower annual proportions of the network in low flow, but this observation does not hold for all years, and the effect is usually small. In summer, the storage capacity has no effect on low flow, except in 2017. The summer of 2017 is close to the median in terms of rainfall but follows a succession of four dry seasons after the summer of 2016. Situations with 280000 m<sup>3</sup> of storage capacity are associated with less summer low flow than situations with 140000 m<sup>3</sup> of storage capacity but still more than in the reference situation. Finally, in autumn, the storage capacity has an influence in 2001, 2003, 2008, 2009, 2012, 2016, 2017, and 2019. For these years, increased storage capacity leads to lower proportions of the network in low flow.

The number of reservoirs consistently affects nearly all the indicators (Figure 5 middle). Only its effect on the autumnal proportion of the network in low flow is inconsistent across years. In general, increasing numbers of reservoirs are associated with higher impacts, i.e., greater decreases in annual and summer outlet discharges and proportions of the network with low flow. With respect to the autumnal proportion of low flow, the effect of the number of reservoirs can occur in either direction depending on the year.

When reservoirs are located more downstream, their hydrological impacts increase, i.e., lower annual and summer discharges at the outlet and higher proportions of the network in low flow throughout the year (Figure 5 right). The only exception occurs in 2002. For this year only, higher numbers of reservoirs and reservoirs located downstream are both associated with lower decreases in annual discharge at the outlet. This finding could be related to the succession of a rainy spring, summer, and autumn in that year.

### 3.2.2 Relative contribution of factors

The boxplots in Figure 5 show that some factors have a stronger effect than others do on the study variables. Figure 6 shows the relative contribution of each factor to the variability of each indicator for each year, according to an analysis of variance. A quick review reveals that (i) the main explanatory factor is different for each indicator (i.e., different main colors on each plot), (ii) for a given indicator, the main explanatory factor can be quite different from year to year (i.e., different color distributions from year to year), and (iii) the residuals of the ANOVAs, i.e., the proportion of variance that is not explained by the study factors, are high for all indicators (i.e., the sky blue color is consistently present throughout the figure).

Storage capacity is the most important factor for explaining the variability in annual outlet discharge, but it has only a limited effect on the variability in summer discharge. For the proportion of the network in low flow, its effect differs depending on the year. In autumn, its contribution to the variance is important in 2001, 2003, 2004, 2008, 2009, 2012, 2016, 2017, and 2019, which corresponds to years with more than 20 % autumnal low flow in the reference situation. In 2001, 2016 and 2017, the storage capacity makes an important contribution to the annual proportion of the network in low flow, corresponding to the three years with the most autumnal low flow in the reference situation. Finally, the storage capacity has a substantial effect on the proportion of low flow in summer only in 2017; it has no effect in the other years.

In most years, the number of reservoirs is clearly the main explanatory factor for the annual and summer proportions of the network in low flow and is consistently followed by the distribution of the reservoirs. The opposite is true for the summer outlet discharge: the distribution of reservoirs is the main explanatory factor, followed by the number of reservoirs. Both factors contribute little and inconsistently to the variance in the annual outlet discharge.

The decompositions of variance for the autumnal proportions of the network in low flow are more difficult to analyze, as the contributions of each factor change every year. If we consider the years with the highest proportions of the network in low

**Figure 5.** Boxplot each of the five indicators of impact (rows) for each simulated year, separating the effect of each factor (columns). The indicators of impact are the absolute change in the annual outlet discharge (a-c), the absolute change in the summer outlet discharge (d-f), the mean annual proportion of the network in low flow (g-i), and the mean proportions of the network in low flow in summer (j-l) and in autumn (m-o). For the proportions of the network in low flow, the large crosses indicate the values in the reference situation.

**Figure 6.** Decomposition of the observed variance in the impact situations for each year and for 5 variables: annual outlet discharge (a), annual proportion of the network in low flow (b), summer outlet discharge (c), and summer (d) and autumnal (e) proportions of the network in low flow. The decompositions of variance are performed with an ANOVA with three explanatory factors and their interactions: reservoir distribution, total storage capacity, and number of reservoirs. The years analyzed span April n to March n+1.

flow in autumn (i.e., more than 20 %), the storage capacity is consistently the main factor, the distribution of the reservoirs is a

405

secondary contributor, and the number of reservoirs has little to no effect.

For all the indicators, the residuals and interaction terms are high and variable from year to year, which means that an important proportion of the variance is not easily explained by our factors (i.e., by a linear model of the modalities of our factors). The indicator with the consistently lowest residuals and interactions is the summer proportion of the network in low flow, but they still represent approximately 25 % of the observed variance. For the summer outlet discharge, they consistently represent at least 40 % of the observed variance. The variability of these terms is the highest for the autumnal proportion of the network in low flow; they can represent 20 % to nearly 100 % of the variance, with a median of 48 %.

### 3.3 The drivers of impacts

In the previous sections, we described the effects of small reservoirs and the effects of the three study factors. Small reservoirs have impacts because they store water that would otherwise flow directly to the outlet and that part of this water can be (i) lost by evaporation, or (ii) withdrawn to irrigate crops. Withdrawals are key, as they determine how much water is taken from the hydrological network and when the reservoirs refill to compensate for the abstractions. Thus, in the following paragraphs, we analyze more precisely the amount and timing of withdrawals in the impacted situations and their consequences on flows.

# 3.3.1 Withdrawals volumes and irrigation return flows

On an annual basis, we can expect that withdrawals in the reservoirs and the decrease in outlet discharge will be strongly linked, provided that the reservoirs are full at the end of the year. The analysis of the absolute change in annual discharge compared with the reference as a function of yearly withdrawals (Figure 7) reveals that (i) withdrawals and decreases in annual outlet discharge are usually higher in situations with the highest storage capacities (280000 m³, blue to pink colors in Figure 7), (ii) the nature and strength of the relationship between both can be quite different from year to year, and (iii) all points align well in a region comprising between 1/2 of withdrawals and 1 of withdrawals for the absolute decrease in outlet discharge. These results indicate that although both variables are linked, annual withdrawals in small reservoirs alone are not sufficient to explain the absolute change in the annual outlet discharge.

In situations with  $140000 \, \mathrm{m}^3$  of storage capacity, withdrawals are always higher than in situations with 3/4 of the capacity threshold (and even higher than the total capacity). This level of withdrawal is possible only if the reservoirs are partially refilled during summer. In situations with  $280000 \, \mathrm{m}^3$  of storage capacity, withdrawals are always lower than the total capacity but sometimes exceed the 3/4 threshold.

The mean interannual values for each simulation (the large colored triangles in Figure 7) align with the line y=3/4x (except for 2 simulated situations). Thus, on average, 3/4 of the irrigation water is used by the plants or is evaporated, and 1/4 returns to the hydrological network. The timing and amount of irrigation return flow can be critical for understanding the effects of small reservoirs, not only on outlet discharges, but also on low flows. These return flows can explain why, for some years (e.g.,

**Figure 7.** Yearly withdrawals in reservoirs compared with the annual absolute change in outlet discharge in m<sup>3</sup> (left y-axis) and in mm (right y-axis) for all situations and all simulated years. Withdrawals and annual outlet discharges are computed from April of year n to March of year n+1. Larger colored triangles correspond to the mean values over the 20-year period for each situation.

2001, 2003, and 2017), the annual proportion of the network in low flow can decrease compared with the reference situation, as this water, which returns a second time to the hydrological network, can increase flows. Hence, irrigation return flows can sometimes sustain flows during dry periods. To explain the variability observed in Figure 7, we can make two assumptions:

- 1. Depending on the year and the situation, the reservoirs can be full or not at the beginning or at the end of the year. Some impacts can thus be deferred from one year to another.
  - 2. Depending on the year (weather events) and the situation (location of irrigable crops and species), the efficiency of irrigation can differ.

Figure 8. Boxplot of the volume needed to reach the maximum storage capacity in reservoirs at the beginning of each season for situations with  $140000 \, m^3$  (plain boxes) and  $280000 \, m^3$  (empty boxes) of total storage capacity. The black horizontal lines indicate 3/4 of total storage capacity. n=45 per box.

# 3.3.2 Timing of withdrawals and reservoir refill

When a reservoir is not full, it collects all the upstream flows, and only the regulatory ecological flow is transmitted downstream, leading to discontinuities in the stream flow. These discontinuities can be critical for understanding the temporal dynamics of reservoir impacts. In our experiment, withdrawals from small reservoirs are highly seasonal. They occur in spring and summer, and most of them are related to the irrigation of maize and soybeans in summer (figure not shown). At the beginning of spring, the small reservoirs are usually full (the volume is close to 0 in Figure 8). There are exceptions each year, when reservoirs are not full at the beginning of spring, which partially confirms the first hypothesis above. In most years, the reservoirs are also full at the beginning of summer. Hence, in spring, the discharges in the hydrological network and the rainfall over the reservoirs are usually high enough to compensate for the withdrawals and evaporation. At the beginning of autumn, the reservoirs are usually not full, meaning that the withdrawals in summer are compensated not only during summer, but also in autumn and sometimes in winter. As a consequence, the stream is disconnected at the location of filling reservoirs during all this time. However, the variability between the years is high, and there are years in which reservoirs are already nearly full at the beginning of autumn

(e.g., 2002, 2005, and 2006). Since stocks and withdrawals are more important in situations with higher storage capacities, the volume to refill after the irrigation campaign is usually much higher in these situations.

### 4 Discussion

In the numerical experiment, we tested the effects of reservoir density, capacity, and distribution along the stream. Given that we tested only one type of reservoir management in one agropedoclimatic context, the results on the impacts of small reservoirs and factor contributions to each indicator should be considered valid only in this specific context of irrigated field crop in catchments dominated by shallow groundwater and clay loam soils. However, the drivers of impacts, i.e., the amount and timing of withdrawals in reservoirs leading to discontinuities in the stream flow and return flows of irrigation, should be the same for other catchments with similar reservoir management, even if they interact differently.

435

430

Our choice of total irrigated surface and irrigable crop distribution is representative of the current situation in southwestern France. The tested values for reservoir density and capacity cover a large variety of plausible situations in the region. It is difficult to estimate whether the distributions of reservoirs in the hydrological network are realistic. In particular, the randomness of the process can lead to situations with many reservoirs on a low-order stream. In the Gélon catchment, there are low-order portions of the network with up to three small reservoirs, so we considered that such configurations are also likely to occur. We were able to test the effects of small dams only. In our context, where groundwater drives most of the flows, hill reservoirs are usually directly connected to shallow groundwater or to drainage water and are most likely to have similar impacts as reservoirs located along the stream.

## 445 4.1 The hydrological impacts of small reservoirs

The main results of the numerical experiment are summarized in Table 2. Except for the autumnal low flow in dry years, small reservoirs have a negative effect on all the indicators. The most severe impacts occur in the summer.

With respect to the impact on annual outlet discharge, our results confirm those of previous works (Habets et al., 2018). For example, Culler (1961) reports a value of -26 %, Tarboton and Schulze (1991) reports a value of -6 %, and Teoh (2003) reports a value of -8 %. Since these values are obtained for different agricultural and hydrological contexts and can represent different quantities (e.g., mean value over a period, worst case value, *etc.*) with possibly different reference states (simulated without reservoirs, measured prior to the construction of reservoirs, *etc.*), the comparison of numerical values seems unhelpful.

Our simulations reveal high interannual variability in (i) the study variables in the reference situation (see Figure 3), (ii) the impacts of reservoirs on these variables (see Figure 4), and (iii) the contribution of each factor to the impacts (see Figure 6). In the literature, simulations are often run on pluriannual time series, yet the question of how the analyses should be performed to

**Table 2.** Summary table of the effect of the three study factors on five indicators of impacts. The first line indicates the global effect ( and ). The other lines indicate the effect and the mean order of importance of each factor in the ANOVA (indicated by the number in brackets). Empty cases indicates that the factor has no impact on the indicator. Low flow refers to the indicator of the proportion of the network in low flow. An increase in low flow is perceived as a negative impact. For the autumnal low flow, only years with more than 20 % of low flow in the reference situation were retained to address the variability in the ANOVA.

|                                                                                                            | Annual<br>outlet<br>discharge    | Summer<br>outlet<br>discharge     | Annual low flow                      | Summer low<br>flow | Autumnal low<br>flow (for dry<br>years) |
|------------------------------------------------------------------------------------------------------------|----------------------------------|-----------------------------------|--------------------------------------|--------------------|-----------------------------------------|
| Global effect of small reservoirs                                                                          | \( \square (-1 to \) \( -6 \% \) | \( \square\) (-20 to \( -60 \% \) | / (for most years)                   | 7                  | V                                       |
| Increasing number of reservoirs                                                                            | ∖ (4)                            | ∖ (3)                             | <b>≯</b> (1)                         | <b>≯</b> (1)       | -                                       |
| $\begin{array}{c} \textbf{Upstream} \rightarrow \textbf{downstream} \\ \textbf{distributions} \end{array}$ | ∖ (3)                            | ∖ (1)                             | <b>≯</b> (2)                         | <b>≯</b> (2)       | <b>≯</b> (3)                            |
| Increasing storage capacity                                                                                | ∖ (1)                            | ∖ (4)                             | -, except for 3 years $\searrow$ (1) | -                  | ∖ (1)                                   |
| Interactions + residuals                                                                                   | (2)                              | (1)                               | (2)                                  | (2)                | (2)                                     |

consider a possibly high interannual variability is never discussed. In most publications, only mean values and/or other statistics over the period are given (e.g., full flow duration curves, Q90, medians, *etc.*) (e.g. in Meigh, 1995; Neal et al., 2002; Schreider et al., 2002; Güntner et al., 2004; Fowler et al., 2015; Pinhati et al., 2020; Robertson et al., 2023). This approach is useful for synthesizing the impacts, but some information on interannual variability is lost. For water managers, the information of interest might not be the mean impacts of reservoirs but the worst-case impacts or the frequency at which the impacts exceed a given threshold. Some authors chose to present their results for specific years only, usually a median year, a dry year, and a wet year (e.g. Tarboton and Schulze, 1991; Teoh, 2003; McMurray, 2006; Deitch et al., 2013; Habets et al., 2014; Lebon et al., 2022). This choice can be good if the climate can be easily classified into typical wet or typical dry years. Our results show that selecting a single dry or a single wet year to analyze the results might be difficult and especially prone to uncertainty, as the prior conditions of a given year can control the impact of that year. The results for dry years may vary depending on whether the previous seasons were dry or wet (see Figure 2). Therefore, we present our results annually for the entire study period and attempt to qualitatively analyze the interannual variability using rainfall information. We could go further and try to find the quantitative links between meteorological variables, the impacts of reservoirs and the effects of the study factors. We also provide no mean values so that readers are presented with the interannual variability of impacts.

Our numerical experiment reveals an important seasonality of impacts. It is particularly important to characterize this seasonality, as the annual impacts of reservoirs on outlet discharge can be considered low (-1 % to -6 %). In absolute and relative terms, the decrease in outlet discharge is highest in summer. It can reach 60-70 %, which is critical for the Gélon stream and for downstream rivers. There are also periods, especially in autumn, in which small reservoirs can lead to an increase in outlet discharge, most likely because of return flows of irrigation. Low flow is also seasonal. Most of it occurs in summer and in autumn in the reference situation, which are the seasons with the greatest impact of small reservoirs. In the literature, the infra-annual variability of reservoir impacts is generally assessed for monthly flows, either using statistics on multiple years (e.g., mean, median, flow duration curves, etc.) (e.g. in Ramireddygari et al., 2000; Savadamuthu, 2002; Alcorn, 2007; Cetin et al., 2009; Habets et al., 2014; Gautam and Corzo, 2023; Yan et al., 2023), or values for specific years (e.g. Tarboton and Schulze, 1991; Thompson, 2012; Dong et al., 2019). In our case, it was not possible to simultaneously analyze the monthly impacts for each year in the 90 simulations. We therefore aggregated our indicators of impact seasonally, an approach that is also sometimes adopted (e.g., in Galéa et al., 2005; Perrin et al., 2012; Xu et al., 2013). Our results on the seasonality of impacts align with what is generally reported in the literature for outlet discharge. When withdrawals are seasonal (e.g., withdrawals for irrigation), then the absolute impact will be more important for some months of the year (e.g. Cetin et al., 2009; Habets et al., 2014; Fowler et al., 2015). Even when withdrawals are constant (e.g., withdrawals for stock watering), the seasonality of stream discharge implies that the relative impact is higher for drier months (e.g. Meigh, 1995; Savadamuthu, 2002; Alcorn, 2007; Robertson et al., 2023). In our case, we observe a combination of both: most withdrawals occur during the drier months. In this work, we focused on low flows, as they are the main concern of local water managers in such agricultural catchments. Low flows have rarely been quantified and analyzed in the context of small farm dams. In the literature, they have been characterized with flow duration curves (Q95 or Q90 as indicators of low flow) (e.g. Hughes and Mantel, 2010; Habets et al., 2014; Pinhati et al., 2020), with the number of days with outlet discharge lower than the historical median discharge (Robertson et al., 2023) or with the minimum mean discharge calculated for a sliding period of 30 days (Galéa et al., 2005). All these indicators are based on outlet discharges, which do not necessarily represent the hydrological state along the entire stream network. Maps of the variables listed above could help locate low-flow hotspots along the stream. Maps are used in some studies on small reservoirs to display the mean discharge over a period (e.g. Güntner et al., 2004; McMurray, 2006; Deitch et al., 2013; Lebon et al., 2022) but never to display information on low flows. However, maps provide only snapshots of the impact for one simulation and one period. They could not be used in this study to compare the 90 simulations. Therefore, our study proposes a novel indicator for the characterization of low flows: the proportion of the network in low flow. It summarizes in one single indicator the hydrological state of the stream network and therefore provides more information than a single observation at the outlet does (see the Supplementary Material for the comparison of low flows at the outlet and for the entire network). Information on hotspot localization is lost, but the values of the indicator can be easily compared between simulations.

Our study reveals that the impact of small reservoirs on the proportion of network in low flow is driven by (i) the decrease in stream discharge downstream of the filling reservoirs and (ii) the return flow from irrigation. Both have contrasting effects and can cause an increase or a decrease in the proportion of the network in low flow depending on the year and the time of the

525

535

540

year (the tendency is an overall increase in low flow). Pinhati et al. (2020) showed that small reservoirs could increase Q95 for some months. In their context (Brazilian savannah), small reservoirs have a reserved flow of at least Q95, even when there is no inflow in the reservoir. Therefore, during dry periods, water is taken from reservoirs to sustain flows (it acts as a "buffer"), which constitutes a notable difference in management compared with our context. All other studies on small reservoirs that focus on low flows have reported either no impact (Galéa et al., 2005) or an increase in low flow (Hughes and Mantel, 2010; Robertson et al., 2023).

### 4.2 The factors underlying reservoir impacts

In this work, we go beyond the evaluation of the impacts of small reservoirs and analyze the influence of factors related to the characteristics of the reservoir network (direction and magnitude). Our main finding is that each of the study factors has a main influence on at least one of the indicators of impact: the storage capacity on the annual outlet discharge, the number of reservoirs on the annual and summer proportions of networks in low flow, and the distribution of reservoirs on the summer outlet discharge. In terms of direction, we found that situations with a higher number of reservoirs and/or with more reservoirs located downstream are usually associated with higher impacts for all the indicators studied. Situations with higher storage capacities are associated with greater impacts on the outlet discharges, but they are also associated with fewer impacts and even greater benefits on the annual and autumnal proportions of the network in low flow for years with a marked low flow in autumn (Table 2).

Many authors have tested scenarios of increased storage capacity (i.e., an increase in the reservoir size or the number of reservoirs) (e.g. Savadamuthu, 2002; Teoh, 2003; Thompson, 2012; Habets et al., 2014). In these publications, increased storage capacity is linked to an increase in water demand in the catchment. The additional water is used to cover the additional demand, which mechanically leads to greater decreases in annual discharge. In our case, we tested situations with different total storage capacities, but the overall water demand determined by the total irrigated area and the distribution of irrigated crops remained the same. Therefore, only the water offer changes. This also results in an increase of withdrawals in situations with higher storage capacities. Few authors have investigated the influence of factors other than storage capacity linked to the spatial characteristics of the reservoir network, i.e., the number of reservoirs at constant capacity and the spatial distribution of reservoirs. Peter et al. (2014) and Ayalew et al. (2015) focused on the influence of reservoir location on flood discharges and flood avalanches. Meigh (1995) tested the effects of reservoirs number and location on mean discharges. However, their modeling approach relies on strong simplifications (e.g., aggregation of small reservoirs into equivalent reservoirs, small reservoir location defined based on drainage area). In particular, withdrawals are constant in time and directly proportional to the total storage capacity. Thus, the number and location of reservoirs only affected the stream discharge through modified losses by evaporation.

In Section 3.3, we stressed that the amount and timing of withdrawals are important as they determine how much water will be abstracted from the stream, when and how much of it will return as irrigation return flows, and when stream disconnections

occur. In the following paragraphs, we provide an interpretation of the effects of the studied factors on the hydrological impacts of small reservoirs with these elements in mind.

The decrease in annual outlet discharge is linked mainly to total water withdrawals in the reservoirs and the proportion of this water that is definitely lost for the stream (i.e., the additional evaporation and transpiration caused by irrigation). Total capacity is a limiting factor for withdrawals in situations with a storage capacity of 140000 m<sup>3</sup>. Thus, it is not surprising that this factor has the greatest effect on annual discharge. In situations and years where the capacity of reservoirs is a limiting factor for withdrawals, situations with more reservoir refill during the cropping season will probably lead to higher withdrawals. The rate of reservoir refilling is probably higher for situations with more reservoirs covering a larger drained area. This point explains the effects of the number and distribution of reservoirs. The proportion of irrigation water that undergoes evapotranspiration depends on the soil water content, crop growth, and weather. It is thus not related to any of our study factors and contributes to the residuals and their interannual variability in the analysis of variance. For some years and some situations (e.g., with many upstream reservoirs), the discharges in autumn and in winter are not sufficient to refill all reservoirs by the end of the year, and the impact on discharges is deferred to the following year, which also increases residues. In summer, withdrawals are important, and the reservoirs are probably refilling throughout the season without reaching their capacity. Therefore, the storage capacity has little effect on the decrease in summer outlet discharge. The proportion of the stream flow produced during summer that will either reach or not reach the outlet and, thus, the amount of area drained by reservoirs are important. The distribution of reservoirs thus is the main factor in the decrease in summer outlet discharge, followed by the number of reservoirs.

The indicator of low flow represents the average length of the hydrological network in low flow. When the reservoirs refill, they create disconnection points on the network, and only the ecological flow is transmitted downstream. Since the low flow threshold and the ecological flow are not computed with the same methodology (Q90 in the reference situation, and 10 % of the mean interannual discharge evaluated with the closest measurement station), the low flow threshold can be higher than the ecological flow. This point explains why the number of reservoirs, i.e., the number of disconnection points, has the greatest effect on the annual and summer low flows. The role of the reservoir distribution can be double. If reservoirs are located downstream, they drain a large area, and they will probably refill more quickly (although it also depends on the number of upstream reservoirs), meaning that the disconnection will last for a shorter period. This situation could occur in autumn, but in summer, we observed that the reservoirs were filling most of the time. However, for downstream RSs, the part of upstream stream flow in the total discharge (composed of the upstream stream flow and the flow from drained surfaces and groundwater units in this RS) will probably be more important, meaning that the discharge will be more affected by a disconnection. Furthermore, in situations where the capacity of reservoirs is a limiting factor for withdrawals, the amount of withdrawals in a season depends on the capacity of the reservoir to refill, and the disconnection can last longer because of higher withdrawals. Given that downstream distributions lead to enhanced annual, summer, and autumnal low flow proportions, this explanation is more realistic. In autumn, the irrigation return flows play an important role in sustaining the flows and decreasing low-flow proportions. For dry years in particular, we observed that situations with higher capacities (i.e., higher withdrawals and return flows) limit the

low flow in autumn and even decrease it compared with the reference situation. We can assume that a higher soil water content at the end of the summer leads to faster soil and groundwater recharge in autumn and thus enhanced baseflows.

Finally, the three study factors alone do not explain the total variability observed in the 90 simulations. For all the indicators and for all the years, the residuals of the ANOVAs are quite important, usually accounting for more than 20 % (Figure 6). These residuals can be qualitatively attributed to complex interactions between reservoirs that are not captured by the three studied factors and to random effects due to the exact localization of irrigable crops and the types of crops irrigated by each reservoir. The second point is important, as we considered that the impacts of reservoirs are caused by the infrastructure and the modification of nearby crops due to the opportunity created by the infrastructure. These elements contribute to the variability in the impacts of reservoirs, which is difficult to analyze. For example, the amount and timing of withdrawals will differ if a reservoir irrigates straw cereals only or maize only. Return flows can have an impact on discharges in the hydrological network, but if they are intercepted by a filling reservoir, their beneficial effects will not be registered downstream. For local water managers, these random effects are impossible to anticipate.

### 5 Conclusions

Our work constitutes a methodological advance and provides new insights into the hydrological impacts of small farm reservoirs and their driving factors. A spatially distributed agro-hydrological model was used to evaluate the impacts of 90 alternative reservoir networks in the same catchment. The model considers crop growth and management at the parcel level and includes a reservoir management model. The networks tested differed in terms of reservoir density, capacity, and spatial distribution, allowing us to study the effects of these factors on flow regimes throughout the year. The focus was on outlet discharges and low flows, for which a new indicator was developed to summarize the hydrological status of the entire hydrological network and not only of its outlet.

Key lessons from the experiments are (i) an important annual variability of impacts and factor influence that needs to be properly analyzed; (ii) the need to consider not only annual but also seasonal indicators given the high subannual variability of the impacts found; (iii) the main explanatory factors to explain the variability in discharges and low flow being alternatively the storage capacity, the number of reservoirs or their spatial distribution; and (iv) the key processes linked with these factors being the amount of timing of irrigation, the disconnections in the hydrological network caused by reservoir refill, and the return flows of irrigation.

We tested the influence of factors related to the characteristics of the reservoir network. A next step will be to test the impact of different management rules. More generally, the approach used in our work for the numerical experiment could easily be applied even in different hydrological and agricultural contexts, provided that models are available for these other managements

https://doi.org/10.5194/egusphere-2025-4737 Preprint. Discussion started: 15 October 2025

and contexts.

Moreover, this work could help support the choice of a representation of small reservoirs in hydrological models depending on the impact indicator relevant for the study, as different properties of the network appear critical to assess different indicators across different timescales. Furthermore, the results of our experiment could be analyzed to test the general assumption made in lumped modeling approaches that the area drained by small reservoirs is a good proxy for their location in the catchment. Finally, our work can also help water managers to better understand the drivers behind the hydrological impacts of small reservoirs and guide them in their decision making.

Code and data availability. Data and code are available from the corresponding author upon reasonable request.

Author contributions. HL: conceptualization, methodology, bibliography, simulations, analysis, visualization, writing; DBL, CD, JM: funding acquisition, supervision, conceptualization, methodology, analysis, review and editing of the paper; CM: analysis, review and editing of the paper

Competing interests. The authors declare that they have no conflicts of interest.

Acknowledgements. This work was funded through the PhD scholarship of HL by the Occitanie Region (France) and the AQUA division of the French National Research Institute for Agriculture, Food and Environment (INRAE). It was also financially supported by the French Biodiversity Agency (Office Français pour la Biodiversité, OFB) through the ESTANH project. The Climae Metaprogram of INRAE and the Key Initiative Water Occitanie (Woc) have also provided in-kind support for this work. We would like to thank David Crevoisier, Armel Thöni, and Dorian Gerardin, the members of the OpenFluid team of the LISAH for their help in performing the modeling work with MHYDAS-Small-Reservoir.

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
