# Peer review of "The influence of small farm reservoir network characteristics on their cumulative hydrological impacts"

_EGUsphere, 2025_

## Author Comment (AC1)

**Response to comments – Referee #1**

This manuscript describes the cumulative impact of small reservoirs on the hydrology of a small catchment area in south-western France. These reservoirs are generated randomly and are connected to the river network. Their characteristics include their water storage capacity, their spatial distribution across the catchment area and their number. The metrics used to assess the impact of these different components are annual flows, summer flows, and the proportion of the hydrographic network at low flow each year. The study is based on 20-year numerical simulations. It shows that the impact of reservoirs is cumulative along the river network and assesses the effects of the distribution, number and capacity of reservoirs on flows, and studies the associated processes. This work is very interesting and also sensitive because it attempts to provide answers to societal questions, in particular the storage of water for irrigation in dedicated reservoirs.

I find this article very well structured and well written. The conclusions are based on objective evidence and clear figures, and the limitations of the study are well presented in the discussion. I recommend publication of this article after minor revisions.

Comments:

- L109: evaporation from the reservoir is considered to be 60% of the reference evaporation, which I assume corresponds to potential evapotranspiration. How valid is this approximation and what is the associated uncertainty? It is important to give an order of magnitude for annual evaporation, as well as annual withdrawals, as these contribute to direct mass loss.

The relationship between reference evapotranspiration (in our case calculated with the Penman-Monteith equation) and actual pond evaporation in MHYDAS-small-reservoir is purely empirical. We adopted a coefficient of 0.6, which is the value most commonly applied in SWAT model applications for simulating actual pond evaporation (Neitsch et al., 2011).

Such empirical relationships are widely used in models representing small reservoirs. However, few studies explicitly report the coefficient value used. Among the 30+ approaches we identified for modeling small farm reservoirs in hydrological models, only 4 give the used value for this parameter (though not all of these approaches model evaporation or use the same method). Jayatilaka et al. (2003) use a value of 0.8 in the model CASCADE, Peter et al. (2014) a value of 0.7 in ResNetM, McMurray (2006) a value of 0.75 (in winter only), and Rabelo (2021) a value of 1.0 in a modified version of SWAT. Note that the three first use the A-pan evapotranspiration as reference.

The reported values vary across studies, which is expected since the climatic context varies between studies. Only, there is limited empirical evidence to guide the selection of an appropriate coefficient. In Southwestern France, no studies are available to justify the choice of one value over another. In a Mediterranean context, where evaporative demand is higher, Martinez-Alvarez et al. (2007) used a modeling approach (energy balance) to derive relationships between A-pan evapotranspiration (measured) and evaporation from irrigation reservoirs (modeled). They found that the pan-coefficient, as they term it, varies throughout the year, with the local weather conditions, and the dimensions of reservoirs. Values are ranging from 0.5-0.6 to 1.2, and are mostly comprised between 0.8 and 1.1. Since the A-pan evaporation is typically higher than the Penman-Monteith evapotranspiration (Allen et al., 1998, Chapter 4), coefficients between Penman-Monteith evapotranspiration and actual evaporation close to or higher than 1 would be justified in this context.

Selecting an appropriate proportionality coefficient is not straightforward. In the case of our numerical experiment, we can assume that the exact value will not have much impacts on the results. Given that the reference evapotranspiration and the actual evaporation are assumed to be proportional, a 50% increase of the coefficient of proportionality would likely result in a 50% increase in actual evaporation (probably a bit less since the surface of the reservoir also decreases). In our experiment, withdrawals are much more important than evaporation, particularly in summer (see supplementary material Figure S6). **Thus, a 50-60% increase in evaporation would have little impact on the outcomes of the numerical experiment. Nonetheless, we acknowledge that our initial choice of 0.6 requires reevaluation in the future, for any context in which the model will be applied.** Reviewer 3 drew our attention to McMahon et al. (2013), which will be a good starting point for this.

To clarify this point for readers, we propose adding the following sentence at line 194 : "With these values for reservoir capacity and irrigable crop surface, reservoirs will be intensively managed. As a result, withdrawals will significantly exceed evaporation losses, which will not be further analyzed in this study (see supplementary material Figure S6 for a comparison of withdrawals and evaporation)."

- L117: withdrawals are possible if the volume of water is greater than 1/4 of the reservoir's capacity: what average water depth does this correspond to, given that the shape is an inverted pyramid? Is this compatible with the characteristics of the withdrawal pumps?

Withdrawals are not permitted when the water volume falls below a minimum threshold. This aims to preserve the pumping equipment and the quality of irrigated water. The threshold was fixed to ¼ of capacity arbitrarily, without considering a specific depth or pump characteristics. This "dead volume" has minimal impact on our results, as tested reservoir capacities were scaled considering that only ¾ of it would be usable.

Given that this point was unclear to the other reviewers, we propose to add the following clarification in L117 : "However, withdrawals from a reservoir are only possible if the water volume exceeds a predefined threshold to preserve the pumping equipment and ensure the quality of irrigated water. In MHYDAS-small-reservoir, the threshold was set at 1/4 of the reservoir's capacity. Since this "dead volume" was accounted for in the design of the numerical experiment, (see 2.2.3),  its exact value is expected to have a limited effects on the study's results".

- L114: on the map of France in Figure 1, there are white pixels in the Rhone valley that should not be there. What do they correspond to?

Thank you for this pointing this out. This is an oversight on our part. It will be corrected in the revised manuscript.

- L127: average annual rainfall is 675 mm: how was this calculated? Using SAFRAN data or measurements from rain gauges located in the catchment area?

This was calculated using SAFRAN data. As part of our field instrumentation on the Gélon catchment, we have a single rain gauge located at the outlet of the catchment. The available record is currently limited to less than 10 years. For the years with available data, there is a good agreement between the measured data and the SAFRAN data.

- L128: The total volume capacity of the reservoirs in the basin is estimated at 205,000 m³. Is this estimate based on the pyramidal shape of the reservoirs or does it come from data describing the various structures?

The data on small reservoirs were obtained from a previous work in a larger area conducted by another laboratory. This study combined multiple databases from various organizations to collect as much information as possible on small reservoirs in the area. Therefore, the data were collected using different methodologies, including field surveys and satellite imagery. For the Gélon catchment, we performed a validation on the field for the 25 reservoirs. As some values of reservoir capacity were obviously incorrect in the database, these were coarsely estimated on the field and corrected.

- L140: What do you mean by "better match expectations"?

In the model used by Lebon et al. (2022), groundwater units were drained only at confluences between two portions of the hydrological network, meaning that the stream discharge between a stream network head and the downstream confluence consisted solely of surface runoff water. In our context of numerous hill aquifers, however, aquifers are drained all along the hydrological network, not only at confluences. The new groundwater units now better capture this continuous groundwater discharge. That is what we meant with "better match expectations".

To clarify this point, we propose to change "but the discharges in the hydrological network better matched expectations" to : "but the new groundwater units better represent a continuous water supply from subsurface hill aquifers along the hydrological network".

- L147: An example of one or two reservoir distributions would be helpful. For example, showing one distribution of the 7, 14 or 21 downstream reservoirs with the two capacities (two colours) as in Figure 1 and the main irrigable crop would help to clarify the ideas.

Examples of reservoir distributions can already be found in the supplementary material (Figures S2 and S3).

Considering this commentary and similar feedback from the other reviewer, we propose moving figure S2 from the supplementary materials to the main article. An improved version of the figure is presented below.

The reference to the supplementary material will be made more explicit so that readers can more easily find figure S3, which provides three examples of networks generated with each of the three methods.

- L180: what is the proportion of reservoirs not connected to the hydrographic network compared to those that are connected? They also contribute to water storage for irrigation.

In the Gélon catchment, 9 reservoirs are not connected to the hydrological network. They have been removed in the numerical experiment as explained in 2.2.4. The addition of the new figure proposed above will also clarify this point.

- L225: a warm-up period of 5 years is considered: does this mean that the simulated data used start in September 2000? Or is the warm-up from 1990 to 1995? Please clarify.

25 years are simulated, and among these 25 years, only 20 can be analyzed. The analysis can start on 2000/09/01. Note that in the result section, only 19 years are analyzed, starting on 2001/04/01 as

it was more relevant to study the impact of reservoirs on a period where they are first emptied and then refilled.

We propose to change "These five years are not included in the analysis" to : "These five years are not included in the analysis. Therefore, the analysis can start in September 2000."

- L233: references to Vidal et al. (2010) and Le Moigne et al. (2020) can be added for SAFRAN. Can you clarify if the reference evapotranspiration is computed from SAFRAN data and at what frequency? Same question for min and max temperatures, do they come from the SAFRAN reanalysis?

For Vidal et al. (2010), we found a reference with the following doi : 10.1002/joc.2003. This does indeed appear to be an important reference, which we will add into the paper (see below our proposed reformulation).

For Le Moigne et al. (2020), we found a reference with the following doi : https://doi.org/10.5194/gmd-13-3925-2020. Although the authors use SAFRAN as a component of their hydrometeorological model, we determined that it does not provide additional relevant information regarding the SAFRAN dataset for our study. Therefore, we have decided not to include this reference.

Thank you for asking about the exact meteorological data used in our study. We realized that we made an error in this regard in the preprint. The data was obtained in mid-summer 2024 with the latest reanalysis. Below are the exact products that we used for each input variable :

Rainfall : hourly values (column Rainf).

T_moy : hourly values (column Tair).

T_min : daily values (column tinf_h_q).

ET0 : daily values (column etp_q, which corresponds to potential evaporation computed with the Penman-Monteith equation).

We propose a new formulation for this paragraph (see our answer to your commentary below).

- L236: You state that meteorological data variability is taken into account: I would temper this statement, as the climate in such a small area, covered by four contiguous points, probably does not vary greatly, at least you have not demonstrated that it does. I suggest removing this idea of variability. However, it is interesting to note that Gélon covers only four cells of the SAFRAN grid. Further on, you only take one cell (8558) into account in your comparison. You could at least show that the annual precipitation for these four grid points is very similar, which would allow you to take only one into account.

Thank you to draw our attention on this point. The statement will be removed. First, we made another mistake, the Gélon only intersects two cells (this will be rectified). Second, more than 97% of the catchment is located on the same cell, that's why we only presented this cell (see Figure 43 in Lebon 2021).

We propose to change this paragraph to : "The weather data are composed of hourly SAFRAN data for rainfall and temperature, and daily data for the minimum daily temperature and reference evapotranspiration (Penman-Monteith). The SAFRAN climatic data are provided by Météo-France and were downloaded via the SICLIMA platform developed by AgroClim-INRAE.  The data are

reanalysis of observed data on 8x8 km2 cells (Bertuzzi et al., 2022; Vidal et al., 2010). The Gélon catchment intersects with 2 of these cells. One of these two covers more than 97 % of the catchment area. The seasonal rainfall for the main cell is presented in Figure 2."

- L252: Figure 2, Medians are extending before 2001 and after 2019: do they use data covering 2000-2020? If not lines must be croped to adjust to 2001-2019.

Medians were computed for the presented period, which will be the period of analysis for the other Figures. This will be modified.

- L300: the sentence 'highlights the role of weather' is not very adapted and accurate: please rephrase

True, we propose to change it for : "which shows that the weather is a determinant factor of reservoir impacts".

- L321: (i) and (v) are the same, (v) is the autumn proportion of the framework in low flow

OK, thanks !

- L327: Fig 5 (a) and (d) respectively

OK, thanks !

- L329: 'the boxes are more separated' is not very adapted, perhaps change it into 'each year, the departure to the median is systematically more pronounced in (a) as compared to (d)'

We propose to change it for: "each year, the intersection between boxes is systematically smaller in (a) as compared to (d)."

- L403: equation should be y=-(3/4)x ; in Figure 7 equations should contain brakets y=-(a/b)x

OK, this will be changed.

Edits:

- L222: parameterization OK, thanks !

- L166: reservoir OK, thanks !

- L176: 1.06 km$^{-2}$ OK, thanks !

- L250: reference simulation

In the paper, we frequently use the reference situation to refer to the scenario without reservoirs, or to refer to the simulation of this reference situation. As we never use the term "reference simulation", we think it would be clearer to keep "reference situation" in L250.

[Figure]

*Figure 1: The four processing steps to generate a network of reservoir with associated irrigable parcels. The generated network correspond to one of the five networks generated with the following parameters : total capacity: 140000 m³; number of reservoirs: 14; Method for reservoir distribution: balanced. This figure will be moved from the supplementary materials (currently Figure S2) to the main text (in pdf quality). Colors have been changed to be more adapted for colorblind persons.*

**References**

Allen, R. G., Pereira, L. S., Raes, D., & Smith, M. (1998). *FAO Irrigation and Drainage Paper No. 56*. Food and agriculture organization of the United Nations.

Bertuzzi, P., Clastre, P., & Aubry, M. (2022). *Information sur les mailles SAFRAN* (Version V1) [Jeu de données]. Recherche Data Gouv. https://doi.org/10.57745/1PDFNL

Jayatilaka, C. J., Sakthivadivel, R., Shinogi, Y., Makin, I. W., & Witharana, P. (2003). A simple water balance modelling approach for determining water availability in an irrigation tank cascade system. *Journal of Hydrology, 273*(1-4), 81-102. https://doi.org/10.1016/S0022-1694(02)00360-8

Lebon, N. (2021). *Modéliser et analyser l'effet cumulé agro-hydrologique des retenues d'eau dans les bassins versants agricoles*. Université de Montpellier.

Lebon, N., Dagès, C., Burger-Leenhardt, D., & Molénat, J. (2022). A new agro-hydrological catchment model to assess the cumulative impact of small reservoirs. *Environmental Modelling & Software, 153*, 105409. https://doi.org/10.1016/j.envsoft.2022.105409

Martínez Alvarez, V., González-Real, M. M., Baille, A., & Martínez, J. M. M. (2007). A novel approach for estimating the pan coefficient of irrigation water reservoirs. *Agricultural Water Management, 92*(1-2), 29-40. https://doi.org/10.1016/j.agwat.2007.04.011

McMahon, T. A., Peel, M. C., Lowe, L., Srikanthan, R., & McVicar, T. R. (2013). Estimating actual, potential, reference crop and pan evaporation using standard meteorological data : A pragmatic synthesis. *Hydrology and Earth System Sciences, 17*(4), 1331-1363. https://doi.org/10.5194/hess-17-1331-2013

McMurray, D. (2006). *Impact of farm dams on streamflow in the Tod River catchment, Eyre Peninsula, South Australia*. Dept. of Water, Land and Biodiversity Conservation.

Neitsch, S. L., Arnold, J. G., Kiniry, J. R., & Williams, J. R. (2011). *Soil and water assessment tool theoretical documentation version 2009*. Texas Water Resources Institute.

Peter, S. J., De Araújo, J. C., Araújo, N. A. M., & Herrmann, H. J. (2014). Flood avalanches in a semiarid basin with a dense reservoir network. *Journal of Hydrology, 512*, 408-420. https://doi.org/10.1016/j.jhydrol.2014.03.001

Rabelo, U. P., Dietrich, J., Costa, A. C., Simshäuser, M. N., Scholz, F. E., Nguyen, V. T., & Lima

Neto, I. E. (2021). Representing a dense network of ponds and reservoirs in a semidistributed dryland catchment model. *Journal of Hydrology, 603,* 127103.

https://doi.org/10.1016/j.jhydrol.2021.127103

Vidal, J.-P., Martin, E., Franchistéguy, L., Baillon, M., & Soubeyroux, J.-M. (2010). A 50-year highresolution atmospheric reanalysis over France with the Safran system. *International Journal*

*of Climatology, 30*(11), 1627-1644. https://doi.org/10.1002/joc.2003

---

## Author Comment (AC2)

**Response to comments – Referee #2**

The article presents an analysis on impacts of the variability of small reservoirs networks configuration in terms of their number, storage capacity and distribution in space. As the authors state, this type of investigation is a new addition to the study of small reservoirs impacts on the water cycle. The article is well written, and the authors described with clarity and detail the configuration tested and the outcomes they achieved. Below I've listed some comments to the authors, mainly clarification of some parts, then some edits to the text.

I suggest the article to be published after minor revision addressing the comments below.

**Comments/revisions**

Line 94: only agricultural and natural surfaces are referenced, is the model not able to represent urban surfaces, or was this a deliberate choice? This could be addressed in the text

Currently, the model cannot represent urban surfaces. Since the urban areas only represent 4.5 % of the total surface (about 0.9 km²), they were assimilated to natural surfaces for model calibration and validation in Lebon et al. (2022).

We propose to add a sentence in L99 : "Urban areas are assimilated to natural surfaces in the model, as they usually cover a small fraction of agricultural catchments".

Line 109-110: I think a small justification for not considering percolation from the bed and walls could be provided here. The authors for example state in another part that the soil is mostly impermeable, this could be an explanation but it is not explicit

Small reservoirs are build to store water for irrigation and to limit water losses by percolation. We assume that  this is achieved by using local material to construct the dam walls ensure the reservoir's impermeability. In the reality, such reservoirs can leak. Since we do not have any estimation of this term in our context, we assume that leakage is negligible.

We propose to add a brief justification in the paper L109 : "Percolation from the reservoir bed to groundwater or through the dam wall is not considered in the model, as leakage is assumed to be negligible in this context of fine textured soils."

Line 117: is this decision backed by some prior knowledge e.g. based on small reservoirs functionality?

Below a given level in the reservoir, pumping becomes more difficult as sediments can damage pumps. The 25 % threshold is arbitrary, and has only a limited effect in the numerical experiment, as the total capacity has been fixed accounting for this "dead volume" (see our answer to Referee #1 on the topic). *A posteriori* and given the limited impact of these restrictions, it would certainly have been clearer to not consider any restrictions, and test the values of 210000 m³ and 105000 m³ for the total capacity in the numerical experiment.

We propose changing the sentence L117 to clarify this point : "However, withdrawals from a reservoir are only possible if the water volume exceeds a predefined threshold to preserve the pumping equipment and ensure the quality of irrigated water. In MHYDAS-small-reservoir, the threshold was set at 1/4 of the reservoir's capacity. Since this "dead volume" was accounted for in

the design of the numerical experiment (see 2.2.3), its exact value is expected to have a limited effect on the study's results".

Line 179-180: the authors could provide more details explaining what are the "specific locations" of the reservoirs

Here, "specific locations" means that hill reservoirs are usually found in locations that are not fully random, but that could be considered as "strategic" in terms of drained area and the surface runoff generated within it. Field observations also indicates that these reservoirs are often positioned to intercept subsurface flow.

We propose adding the following clarification in the sentence : "As hill reservoirs are usually found in locations where surface and subsurface flow converge, which is not captured by the model, a random placement would not be meaningful for this type of reservoir".

Line 180: at Line 96, the hydrological network is defined as the RSs. In Figure 1, REs are not all located on RSs. This is in conflict with the sentence at line 180-1.

Figure 1 shows the current distribution of reservoirs in the Gélon catchment, with both hill reservoirs and small dams. In the numerical experiment, we remove all reservoirs before generating a new reservoir network, with connected reservoirs only. See the Supplementary material Figure S2 and S3 for examples of generated reservoir networks.

To improve the clarity of the presentation of the numerical experiment, we propose moving figure S2 from the supplementary material to the main text. An improved version of the figure is presented at this end of this document.

Line 221: the authors could state here at what time steps the simulations are performed

In line 92, we precise that the model runs at an hourly time step for water routing and a daily time step for crop growth. According to us, it is not necessary to repeat it in line 221.

Line 298: the authors introduce here "irrigation return flows", which are later addressed in a dedicated section, but it is not clear to me what these flows are or why they happen, are they a component of the model? How and why does it return to the hydrological network? The reservoirs are emptied at the end of the irrigation season? Is this happening in reality?

Irrigation water entering the soil increases soil water content. Soil water can either (i) be transpired by plants, (ii) evaporate to the atmosphere, (iii) percolate to groundwater. At harvest, not all irrigation water has necessarily been used by plants or evaporated. Some of it either leached to groundwater, or still remains in the soil, and will eventually percolate later once the soil reaches saturation after subsequent rain events. These percolation fluxes that would not happen without irrigation increase groundwater stocks and therefore streamflows. Such additional flows, generated by irrigation, are referred to as irrigation return flows. They represent water losses from an agricultural standpoint, and are often studied in the context of nutrient and pesticides leaching (Causapé et al., 2006; Poch-Massegú et al., 2014).

For more consistency in the result section, we propose :

- Removing the introduction of irrigation return flow in L298. This paragraph (3.1.2) aims at describing the observed impacts. The statement on irrigation return flow is an interpretation and does not belong here.

- Defining the irrigation return flows in the dedicated section (3.3.1 Withdrawals volumes and irrigation return flows), in L405 : "Thus, on average, 3/4 of the irrigation water is used by the plants or evaporated. The remaining ¼ returned to the hydrological network as irrigation return flow, defined as the portion of irrigation water that flows back to the hydrological network (Poch-Massegú et al., 2014). Irrigation increases soil water content. The applied water can therefore contribute to different fluxes, i.e. (i) crop transpiration, (ii) soil evaporation, and (iii) percolation. Irrigation return flows occur when part of the irrigated water percolates to groundwater, which increases water table levels and streamflows. Since percolation only occurs when soils water content is above field capicity, irrigation return flows can be delayed compared to the irrigation period. The timing and amount of irrigation return flow can be critical for understanding the effects of small reservoirs, not only on outlet discharges, but also on low flows. These return flows can explain why, for some years, the autumnal outlet discharge increases compared to the reference situation (e.g. 2002, 2005, 2006, 2011, 2017). These return flows occur at reach sections that are located near irrigated fields, and can locally sustain flows during dry period. That explains why the proportion of network in low-flow decreases for some years, especially in autumn (e.g. 2001, 2016, 2017). To explain the variability observed ..."

Line 391-392: withdrawals are based on the decision model. Can observation (i) be proposed as an "absolute" observation as it is proposed here?

Figure 7 shows that the decrease in outlet discharge is linked with the amount of withdrawals. This result is expected. On the Figure, we see that withdrawals are higher in situations with higher storage capacities (with equal irrigable surface), which was also expected since the lowest value of capacity was design to be insufficient to cover all water demand. In a real case, we can expect that farmers with larger reservoirs will also tend to irrigate more by expanding irrigated areas or increasing irrigation rates, provided that reservoirs can be fully recharged during the wet season. Our list of observations starting in L391 only describes and helps interpret Figure 7 and are specific to our context. At the beginning of the discussion (L429), we stress that our results are to be interpreted considering the study context, i.e. the context of "irrigated field crop in catchments dominated by shallow groundwater and clay loam soils", which also includes the assumptions regarding farmer water use embedded in the decision model. We opened the discussion with these statements to stress that the impacts of small reservoirs inherently depend on their management and usage, which should be explicitly addressed in this type of work. In our case, we specify the management rules in paragraphs 2.1.2 and 2.1.3 (i.e intense use of water when available). Additionally, paragraph 2.2.3 specifies that reservoir capacities were chosen based on the mean yearly irrigation water demand, meaning that the stored volume will not always be sufficient to cover all needs, especially in situations with 140000 m³ of storage capacity.

**Edits**

Line 13: here, "low flow proportions" is not too clear, I would change to "the proportion of low flow"

Ok, it will be done in the revised version.

Line 14: would change to "For the two indicators", since they are presented a few sentences before

Ok, it will be done in the revised version.

Line 43-44: reference to Colombo et al., 2024 can be done as a more recent example of small reservoirs cumulative impacts study through modeling (https://doi.org/10.1016/j.jhydrol.2024.130640)

Ok, it will be done in the revised version.

Line 78: streamflow

Ok, it will be done in the revised version.

Line 276: i would suggest to put (no reservoirs) after "reference situation"

Ok, it will be done in the revised version.

Line 418: I would remove "figure not shown"

Ok, it will be done in the revised version.

Table 2 caption: where the upward and downward arrows are indicated I would add text stating something like "increase" and "decrease", as it could be misunderstood as "positive" and "negative"

Ok, it will be done in the revised version.

[Figure]

*Figure 1: The four processing steps to generate a network of reservoir with associated irrigable parcels. The generated network correspond to one of the five networks generated with the following parameters : total capacity: 140000 m³; number of reservoirs: 14; Method for reservoir distribution: balanced. This figure will be moved from the supplementary materials (currently Figure S2) to the main text (in pdf quality). Colors have been changed to be more adapted for colorblind persons.*

**References**

Causapé, J., Quílez, D., & Aragüés, R. (2006). Irrigation Efficiency and Quality of Irrigation Return Flows in the Ebro River Basin : An Overview. *Environmental Monitoring and Assessment*, *117*(1), 451-461. https://doi.org/10.1007/s10661-006-0763-8

Lebon, N., Dagès, C., Burger-Leenhardt, D., & Molénat, J. (2022). A new agro-hydrological catchment model to assess the cumulative impact of small reservoirs. *Environmental Modelling & Software*, *153*, 105409. https://doi.org/10.1016/j.envsoft.2022.105409

Poch-Massegú, R., Jiménez-Martínez, J., Wallis, K. J., Ramírez de Cartagena, F., & Candela, L. (2014). Irrigation return flow and nitrate leaching under different crops and irrigation methods in Western Mediterranean weather conditions. *Agricultural Water Management*, *134*, 1-13. https://doi.org/10.1016/j.agwat.2013.11.017

---

## Author Comment (AC3)

**Response to comments – Referee #3**

This manuscript examines the impact of small dams on the flow regime in a small catchment in south western France. The aim of the study appears to be to systematically assess how the spatial arrangements and physical characteristics of dams affect their hydrological impact.

The study uses a series of randomly generated arrangements of dams placed on the stream network with randomly generated physical characteristics. While many studies tend to focus on annual flows, this study assesses additional flow metrics including seasonal flows and the proportion of the stream network experiencing low flows. Notably, it models the effect of irrigation return flows which is novel. This appears to have a significant effect on results.

The results highlight some interesting relationships between the hydrological impacts and the number, volume, and arrangement of dams. However, there are a small number of assumptions which limit the applicability, including the use of water for irrigation with return flows, passing environmental flows, and limiting withdrawals to the top 75% of dam capacity. Modifying these parameters (perhaps in future studies) could provide results which are relevant to many other contexts and jurisdictions.

The article is well written, easy to understand, and clearly structured. The discussion and results are relevant for the fields of water resources management and hydrology. I am keen to see this paper published, however as noted by other reviewers, it could be improved with some minor revisions.

Specific comments:

L109 – Evaporation from the surface of each dam is assumed to be equal to 60% of ET. This may be appropriate, but it would be good to see some justification for this. Other evaporation variables may be more suitable for this purpose, such as Morton's shallow lake evaporation (refer to doi 10.5194/hess-17-1331-2013).

We used the value of 0.6, which is the one most commonly applied in SWAT model applications for simulating actual pond evaporation. Our approach to modeling pond evaporation is empirical, and while it is widely used in hydrological models to represent small reservoir evaporation, few studies report the coefficient values used. We acknowledge that the representation of evaporation could be improved in our model, and we thank you for providing the useful reference.

In our experiment, withdrawals is the main loss term on reservoirs (see Supplementary Figure S6). Consequently, a change in reservoir evaporation would probably have only a limited impact on our results. This justifies our decision not to focus further on refining our approach to pond evaporation yet.

We propose adding the following sentence at line 194 (same proposal as for Reviewer #1 who raised a similar question) : "With these values for reservoir capacity and irrigable crop surface, reservoirs will be intensively managed.  As a result, withdrawals will significantly exceed evaporation losses, which will not be further analyzed in this study (see supplementary material Figure S6 for a comparison of withdrawals and evaporation)."

L114 – In the results section, many references are made to return flows from irrigation. This feature of the model and assumptions around it are not described anywhere in the method (eg. percent of flow returned, which reach does irrigation water return to). I would expect to find this information

in Section 2.1.3. I realise this information may be present in one of the references given, but it appears to be important for this study so it needs to be provided here.

This point was also raised by Reviewer #2. In our paper, we incorrectly assumed that the concept of irrigation return flows was widely understood. This term refers to the portion of irrigated water that is neither evaporated nor transpired, but instead percolates to groundwater and consequently may further re-enters the hydrological network. Our model does not assume a fixed proportion of irrigation return flows for a given irrigation height. These are computed as part of the percolation from the soil water balance in the soil-crop model. *A posteriori,* we observe that streamflow increases in late summer and autumn in situations with irrigation compared to the non-irrigated reference situation. We attribute this increase to irrigation return flows. By comparing the change in outlet discharge and the total withdrawals in reservoirs, we can estimate these return flow. If all withdrawn water for irrigation was evaporated or transpired, the net decrease in outlet discharge would equal the total withdrawals. Figure 7 of the paper shows that the net decrease ranges between ½ and 1 times the withdrawals,with the difference representing irrigation return flows.

For more consistency in the result section, we propose (same proposal as for Reviewer #2) :

- Removing the introduction of irrigation return flow in L298. This paragraph (3.1.2) aims at describing the observed impacts. The statement on irrigation return flow is an interpretation and does not belong here.

- Defining the irrigation return flows in the dedicated section (3.3.1 Withdrawals volumes and irrigation return flows), in L405 : "Thus, on average, 3/4 of the irrigation water is used by the plants or evaporated. The remaining ¼ returned to the hydrological network as irrigation return flow, defined as the portion of irrigation water that flows back to the hydrological network (Poch-Massegú et al., 2014). Irrigation increases soil water content. The applied water can therefore contribute to different fluxes, i.e. (i) crop transpiration, (ii) soil evaporation, and (iii) percolation. Irrigation return flows occur when part of the irrigated water percolates to groundwater, which increases water table levels and streamflows. Since percolation only occurs when soils water content is above field capicity, irrigation return flows can be delayed compared to the irrigation period. The timing and amount of irrigation return flow can be critical for understanding the effects of small reservoirs, not only on outlet discharges, but also on low flows. These return flows can explain why, for some years, the autumnal outlet discharge increases compared to the reference situation (e.g. 2002, 2005, 2006, 2011, 2017). These return flows occur at reach sections that are located near irrigated fields, and can locally sustain flows during dry period. That explains why the proportion of network in low-flow decreases for some years, especially in autumn (e.g. 2001, 2016, 2017). To explain the variability observed ..."

L117 – As others have noted, assuming no withdrawals if the dams is below 25% capacity seems arbitrary. Is this a regulated limit? Or standard operating procedure? In most parts of the world, farmers will extract water from their dams until it is just a small pool of mud. A 10 word explanation is fine.

As your comment is related to a comment of Reviewer #1, we will provide the same response.

Withdrawals are not permitted when the water volume falls below a minimum threshold. This aims to preserve the pumping equipment and the quality of irrigated water. The threshold was fixed to ¼ of capacity arbitrarily, without considering a specific depth or pump characteristics. This "dead

volume" has minimal impact on our results, as tested reservoir capacities were scaled considering that only ¾ of it would be usable.

As suggested, we propose adding the following clarification in L117 : "However, withdrawals from a reservoir are only possible if the water volume exceeds a predefined threshold to preserve the pumping equipment and ensure the quality of irrigated water. In MHYDAS-small-reservoir, the threshold was set at 1/4 of the reservoir's capacity. Since this "dead volume" was accounted for in the design of the numerical experiment, (see 2.2.3), its exact value is expected to have a limited effects on the study's results".

L155 – I do not quite understand the scenarios. In particular, the phrase 'each combination of modalities is repeated 5 times' is not clear. Exactly what is repeated 5 times? And what is different about each of these 5 scenarios?

The key point of our numerical experiment is the random generation of reservoir networks. Our network generator requires three parameters : the number of reservoirs, the total capacity to be distributed between these reservoirs, and a parameter controlling the random spatial distribution of reservoirs (upstream vs downstream). The experimental plan consists in testing three different values for the number of reservoirs, two for the total capacity, and three for the distribution of reservoirs. Since network generation is stochastic, we performed five repetitions for each combination of parameters, resulting in a total of 90 generated networks and 90 simulations ($3 \times 2 \times 3 \times 5$).

As the network generation process was unclear to another other reviewer also, we propose moving figure S2 from the supplementary material to the main article. An improved version of the figure is presented at this end of this document.

Table 2 – Typically, I would expect dams to have effects throughout summer and into autumn while they are filling. This extends the summer low flow season further into autumn. Some results here seem to show slightly different autumn results where irrigation return flows are significant. An interesting follow up study could be to repeat the study with water withdrawals for stock or domestic use with no irrigation returns, or without the 10% environmental flow requirement which is not required in many other jurisdictions. The results may appear quite different. I recommend the paper briefly recognises that these results are dependant on some assumptions which may not always be applicable in other locations.

You're right. As is the case for any study on small farm reservoirs, our results are context dependent. Throughout the paper, we tried to clearly present our context and assumptions. At the beginning of the discussion (L429 to L444), we give the domain of validity of the results, the "specific context of irrigated field crop in catchments dominated by shallow groundwater and clay loam soils".

As Table 2 summarizes our result, we propose adding in the caption a statement to reaffirm that these results are context dependent : "Table 2. Summary table of the effect of the three study factors on five indicators of impacts **found in the numerical experiment. These results are valid in the context of the study.** The first line indicates the global effect (     and     ). The other lines indicate the effect and the mean order of importance of each factor in the ANOVA (indicated by the number in brackets). Empty cases indicates that the factor has no impact on the indicator. Low flow refers to the indicator of the proportion of the network in low flow. An increase in low flow is perceived as a

negative impact. For the autumnal low flow, only years with more than 20 % of low flow in the reference situation were retained to address the variability in the ANOVA. "

As explained above, we cannot turn on or turn off the irrigation return flows. Their importance may vary in other contexts (e.g. deeper soils, different hydrological functioning, more conservative irrigation).
Please note that this article is part of the PhD work of Henri Lechevallier. The question of reservoir management and especially the connection to the stream appears central to us also. Therefore, it will be the focus of a second numerical experiment, which we intend to publish in the future.

[Figure]

*Figure 1: The four processing steps to generate a network of reservoir with associated irrigable parcels. The generated network correspond to one of the five networks generated with the following parameters : total capacity: 140000 m³; number of reservoirs: 14; Method for reservoir distribution: balanced. This figure will be moved from the supplementary materials (currently Figure S2) to the main text (in pdf quality). Colors have been changed to be more adapted for colorblind persons.*